



# Assessing the stability of LSTM runoff projections in Switzerland under climate scenarios

Fabien Courvoisier[1], Basil Kraft[1], Yann Yasser Haddad[1], Massimiliano Zappa[2], and Lukas Gudmundsson[1]

[1]Institute for Atmospheric and Climate Science, ETH Zurich, Zurich, Switzerland
[2]Swiss Federal Institute for Forest, Snow and Landscape Research WSL, Birmensdorf, Switzerland

**Correspondence:** Fabien Courvoisier (fcourvoisier@ethz.ch)

**Abstract.** Climate change is intensifying the global water cycle, altering both mean runoff and extremes, and strengthening the need for reliable hydrological projections to support adaptation. Traditionally, such projections have relied on process-based models. More recently, machine learning models, and in particular Long Short-Term Memory (LSTM) networks, have shown strong skill in predicting and reconstructing runoff from observations, raising interest in their use for hydrological projections.

However, their ability to provide stable and physically credible results when forced with future climates beyond their training domain remains largely unexplored. Here we evaluate this question in Switzerland, a region strongly exposed to warming due to its alpine environment and glacier influence. An LSTM trained on observed meteorological and discharge data is driven with CH2018 climate and glacier projections for 1981–2100, and benchmarked against Hydro-CH2018 simulations from the process-based model PREVAH under identical forcings. Results show that the LSTM reproduces key hydrological signals

closely – wetter winters, drier summers, and elevation-dependent trends – consistently across catchments and climate chains. Divergences are most pronounced in alpine and glacier-fed catchments, where runoff dynamics are more complex, yet the main governing patterns are captured. The largest limitation arises for extremes, where the LSTM underestimates peak flows, consistent with previously reported saturation effects. Overall, this study demonstrates that LSTMs can deliver robust mean-flow projections and trends comparable to a process-based benchmark, while highlighting persistent challenges in representing

hydrological extremes.

## 1 Introduction

Climate change is intensifying the global water cycle, altering both average runoff conditions and the frequency of hydrological extremes (Seneviratne et al., 2021; Hock et al., 2019; Gudmundsson et al., 2021). These impacts are not uniform across regions,

with mountainous areas being particularly sensitive: warming reduces snow storage and glacier mass, leading to earlier melt and diminished summer runoff (Seneviratne et al., 2021; Hock et al., 2019; Kotlarski et al., 2022; Brunner et al., 2019b). Such regime shifts also propagate downstream, affecting non-mountainous areas. The European Alps, and Switzerland in particular,



exemplify these dynamics, with projections consistently showing wetter winters, drier summers, both in precipitation and runoff, and elevation-dependent hydrological responses (FOEN, 2021; Kotlarski et al., 2022; Brunner et al., 2019a, b). These

changes have far-reaching implications for both human sectors and natural ecosystems dependent on stable water supply, including hydropower, agriculture, and natural hazard management. (Hock et al., 2019; Seneviratne et al., 2021; Brunner et al., 2019a; Haddad et al., 2025). Robust runoff projections are therefore essential for long-term planning and adaptation (Kotlarski et al., 2022; Brunner et al., 2019a).

In recent years, machine learning (ML) models, and particularly Long Short-Term Memory (LSTM) networks, have emerged

as powerful tools for rainfall–runoff modelling, because they can learn non-linear relationships, long-term dependencies, and storages directly from data, while allowing rapid inference once trained (Hochreiter and Schmidhuber, 1997; Kratzert et al., 2018; Clark et al., 2024; Kraft et al., 2024). Large-sample evaluations demonstrate that LSTMs match or surpass conceptual models across diverse conditions: they outperformed the Sacramento Soil Moisture Accounting model in snow-influenced US basins (Kratzert et al., 2018), exceeded lumped models in a Chinese basin (Man et al., 2023), improved NSE in 69 %

of Australian catchments while generalizing better to unseen climate extremes (Clark et al., 2024), and added ≈0.10 NSE over four national models in Great Britain, proving reliable even in catchments considered difficult to model (Lees et al., 2021). These results highlight their robustness and scalability; with sufficiently broad training data, a single LSTM captures diverse hydrological behaviors and achieve strong generalization performance (Slater et al., 2024; Frame et al., 2022). As a consequence, LSTMs are now considered as candidates for next-generation operational hydrological modelling (Clark et al.,

2024; Kratzert et al., 2019a; Nearing et al., 2024).

Despite their success in present-day forecasting, only a handful of studies have tested LSTMs for multi-decadal climate projections. Lee et al. (2020) found that an LSTM reproduced historical Mekong runoff and projected future trends similar to a calibrated SWAT model. In contrast, Song et al. (2022) reported good historical skill in South Korea's Yeongsan basin, yet the LSTM projected a 13 % runoff decrease under SSP5-8.5 while SWAT projected an 18 % increase, illustrating divergence

under high-emission forcing. Using synthetic climate perturbations, Martel et al. (2024) found that an LSTM agreed with four conceptual models on the overall magnitude of projected runoff changes, but differed in its sensitivity to climatic inputs, pointing to a different representation of runoff-climate relationships. Similarly, Natel De Moura et al. (2022) observed that an LSTM's skill dropped when tested on unseen wet/dry regimes in Swiss catchments, but recovered to bucket-model levels when the training set spanned a broader climatic range.

Collectively, these studies confirm that LSTMs can replicate process models in some contexts yet may diverge when forced outside their calibration climate. Standard LSTMs are purely data-driven and impose no physical constraints, making them vulnerable to unrealistic behaviour when exposed to novel climate inputs (De Silva et al., 2020; Read et al., 2019; Shortridge et al., 2016; ElGhawi et al., 2025). For instance, Wi and Steinschneider (2022, 2024) demonstrated that a regional LSTM trained on Californian data projected runoff increases under warming, contrary to evapotranspiration theory, unless retrained

with more diverse climates or supplied with physically based ET inputs. Similarly, Baste et al. (2025) stress-tested an LSTM with synthetic precipitation extremes and found that discharge predictions could not reproduce extreme values from training data and plateaued even before rainfall exceeded the historical maximum. Diagnostics traced this to gate-activation saturation,



which prevented the network from scaling its response to unprecedented inputs and systematically underestimated peak flows. Beyond this saturation issue, predicting extremes remains inherently difficult for neural networks and other ML models, as they require large training datasets while extreme events are, by definition, rare (Martel et al., 2024). Thus, although LSTMs excel in retrospective hydrology or short-term forecasting, ensuring stability and physical credibility under out-of-distribution climate forcing remains an important and open challenge in the field.

While concerns remain about the robustness of LSTM models under changing climate conditions, some studies suggest that when trained on large and diverse datasets, these models can implicitly learn runoff-relevant physical relationships and produce out-of-sample simulations that rival calibrated process-based models (Kratzert et al., 2019a). Building on this hypothesis, the present study investigates whether an LSTM model trained on diverse observed meteorological and discharge data can generate stable and physically plausible runoff projections when exposed to out-of-distribution inputs from the Swiss CH2018 climate scenario ensemble (CH2018, 2018).

To this end, we adopt the architecture and training setup developed by Kraft et al. (2024), which demonstrated high accuracy in reconstructing runoff across Swiss catchments, and extend it by driving the trained LSTM with an ensemble of RCP8.5 climate and glacier projections for the period 1981–2099. The resulting LSTM-based projections are benchmarked against outputs from Hydro-CH2018, a national-scale simulation dataset generated using the process-based hydrological model PREVAH (Precipitation-Runoff-Evapotranspiration HRU Model; Viviroli et al. (2009)), forced with the same CH2018 regional climate model (RCM) ensemble (Brunner et al., 2019a).

This comparison enables us to assess (i) the temporal and spatial coherence of LSTM versus PREVAH runoff for both the historical baseline and future periods, (ii) the ability of the LSTM to reproduce key Alpine hydro-climatic signals – winter increases, summer drying, and elevation-dependent trends – across Switzerland's representative pluvial, nival, and glacio-nival regimes, and (iii) model behaviour for flow extremes, with particular attention to potential LSTM saturation effect. To our knowledge, this is the first national-scale study to evaluate the long-term stability of an observation-trained LSTM, dynamically coupled to glacier evolution, when driven by the CH2018 RCM ensemble and benchmarked side by side with a conceptual model across the entire 1981–2100 period.

## 2 Data

### 2.1 Observed runoff

We used the same observed runoff dataset as in Kraft et al. (2024), sourced from the CAMELS-CH database (Catchment Attributes and Meteorology for Large-sample Studies – Switzerland) (Höge et al., 2023). Runoff observations are provided at daily resolution by hydrological gauging stations installed at the outlets of Swiss catchments. Most stations are operated by the Swiss Federal Office for the Environment (FOEN, 2024), with additional contributions from cantonal agencies in Aargau, Baselland, Bern, St. Gallen, and Zurich. The full dataset comprises 267 gauging stations and spans the period 1961–2024. For model training, we selected a subset of 96 catchments with minimal anthropogenic influence (e.g., hydropower, reservoirs), based on CAMELS-CH metadata and the classification of Brunner et al. (2019c).





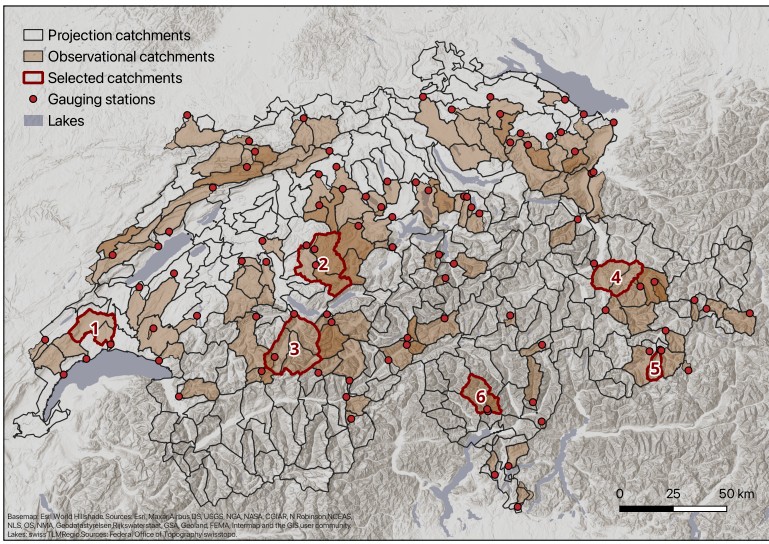

**Figure 1.** Map of Switzerland showing catchment domains. Brown shading indicates the 96 minimally impacted observational catchments (used for LSTM cross-validation scheme), with their gauging stations marked by red dots (identical to Kraft et al. (2024)). Grey outlines denote the 307 projection catchments, identical to those comprising the HydroCH2018 dataset (Brunner et al., 2019a). Red outlined catchments indicate the catchments selected for the second part of our analysis, selected following Muelchi et al. (2021a). Catchment boundaries are initially taken from the Swiss Federal Office for the Environment (BAFU, 2016).

## 2.2 Observed atmospheric forcing

Daily precipitation and temperature data were derived from high-resolution, gridded observational products produced by MeteoSwiss: RhiresD for precipitation and TabsD for mean temperature (MeteoSwiss, 2021a, b). Both datasets start in 1961 and provide daily coverage across Switzerland.

RhiresD combines rain-gauge data from ∼430 Swiss stations, supplemented by stations in neighbouring countries, through a two-step interpolation procedure: first, a PRISM-based regression generates a climatological precipitation field for the 1971–1990 baseline. Daily station anomalies are then interpolated using the SYMAP algorithm and applied to the climatology to reconstruct daily fields. TabsD, in contrast, is based on about 90 long-term SwissMetNet stations, with additional high-elevation series to improve spatial accuracy. Its interpolation relies on a non-linear vertical profile and non-Euclidean distance weighting, enhancing performance in alpine terrain.

Both RhiresD and TabsD are widely used in hydrological and climate modelling applications (Kraft et al., 2024). For model training in this study, daily precipitation and temperature were spatially averaged over the 96 selected catchments, aligning with the catchment delineations.





## 2.3 Catchment properties

Static catchment descriptors were incorporated during training to improve generalization across catchments and to inform the LSTM model about physiographic differences between them (Kraft et al., 2024). The dataset includes most of the variables used in Kraft et al. (2024) and combines those typically employed to force the spatially distributed model PREVAH.

The descriptors fall into four categories:

- Topography: elevation and slope, derived from MERIT DEM (Yamazaki et al., 2017) and complemented with swissALTI3D
for improved accuracy in Alpine terrain (Swisstopo, 2024; Weidmann et al., 2018).

- Land use: reclassified CORINE Land Cover data (European Environment Agency, 2018), aggregated into seven categories (artificial, agriculture, forest, shrubs, barren, wetlands, water) and complemented with the Swiss habitat map for national consistency (Price et al., 2023).

- Soil properties: extracted from the SoilGrids database, a global product based on machine learning models trained on
soil profiles (Hengl et al., 2017; Poggio et al., 2021), and enriched with high-resolution Swiss forest data (Baltensweiler et al., 2022). Variables include bulk density, sand, silt, clay content, and coarse fragment volume for the 0–30 cm layer, spatially averaged per catchment.

- Catchment area: derived from the delineated CAMELS-CH catchment polygons (Höge et al., 2023).

All variables were aggregated at the catchment scale and compiled into a single feature matrix for model training, following
the pre-processing workflow of Kraft et al. (2024).

## 2.4 CH2018 climate simulations

The climate projections used in this study stem from CH2018, Switzerland's official climate scenario initiative led by the National Centre for Climate Services (NCCS) (CH2018, 2018; Fischer et al., 2022). We employ the DAILY-GRIDDED product, which provides transient daily temperature and precipitation fields at 2 km resolution for 1981–2099.

CH2018 is based on EURO-CORDEX simulations that cover a historical period forced by observed radiative drivers up to 2005, followed by the corresponding scenario run (RCPs) from 2006 to 2099, yielding continuous transient time series.

The underlying EURO-CORDEX simulations combine a driving Global Climate Model (GCM) that simulates large-scale climate processes at ∼200 km resolution, a Regional Climate Model (RCM) that downscales GCM output to 12 or 50 km over Europe, and one of three emission scenarios (RCP2.6, RCP4.5, RCP8.5).

The CH2018 DAILY-GRIDDED dataset is based on 68 such GCM–RCM–RCP model chains; in this study we use the RCP8.5 subset (N = 14, see Table A1), which provides the strongest forcing signal for evaluating LSTM robustness.

The raw model output is bias-corrected using quantile mapping (QM), calibrated over the reference period 1981–2010 (CH2018, 2018; Maraun et al., 2017; Gudmundsson et al., 2012). Correction is applied individually to each variable using Swiss gridded observations, such that during the calibration period (1981–2010) the statistical distribution of the corrected



daily values approximately matches that of the observations. The resulting transient projections are widely used and well
suited for regional to catchment-scale impact studies (Muelchi et al., 2021b; Brunner et al., 2019a). Further methodological
details are available in the CH2018 Technical Report (CH2018, 2018).

## 2.5    Dynamic glacier fraction

In contrast to Kraft et al. (2024), who assumed static glacier coverage, we incorporate time-varying glacier fraction to better
represent changing hydrological conditions under climate change. The same glacier projections were used in Hydro-CH2018.
The glacier projections used in this study are based on simulations conducted by Brunner et al. (2019b), who applied the
methodology of Zekollari et al. (2019) to estimate glacier evolution across Switzerland.

    Glacier evolution was simulated using the GloGEMflow model (Zekollari et al., 2019), which couples a temperature-index
surface mass balance (SMB) model, driven by E-OBS observational climate data (1951–2017) and EURO-CORDEX RCM-
based projections, with a flowline ice dynamics module based on the shallow-ice approximation (SIA). The model was cali-
brated using geodetic mass balance data from the World Glacier Monitoring Service (WGMS) and initialized in 1990 under a
steady-state assumption (Brunner et al., 2019b).

    Simulations were conducted for 30 CH2018 model chains, producing gridded glacier area maps at 5-year time steps from
1990 to 2100. This dataset was then used as to compute catchment-level annual glacier area fractions over time, which served
as input for our LSTM model setup (See Section 3.4 for preprocessing details).

## 2.6    Benchmark: Hydro-CH2018

The Hydro-CH2018 dataset provides simulated daily runoff for 307 medium-sized Swiss catchments from 1981 to 2100 (Brun-
ner et al., 2019a). It was generated with the semi-distributed conceptual hydrological model PREVAH, developed for mesoscale
applications in mountainous regions (Viviroli et al., 2009). PREVAH simulates the water cycle through process modules for
interception, evapotranspiration, soil water dynamics, snow and glacier melt, groundwater flow, runoff generation, and routing.

    In Hydro-CH2018, PREVAH was implemented on a $500 \times 500\,\mathrm{m}$ computational grid, providing sufficient spatial resolution
to capture runoff variability across Swiss catchments. Simulations were driven by observed meteorological inputs for the
historical period (1981–2005) and by downscaled, bias-corrected CH2018 climate chains (39 GCM–RCM combinations) for
the projection period (1981–2100), including observation-driven reference runs (1981–2005) and scenario-driven projections
(2006–2100). The forcing variables comprised daily precipitation, temperature, global radiation, relative humidity, and wind
speed, with temperature lapse-rate correction.

    Future glacier evolution was simulated externally and integrated into the hydrological modelling to represent glacier runoff
contributions, using GloGEM (Huss and Hock, 2015) for smaller glaciers and GloGEMflow (Zekollari et al., 2019) for larger
glaciers. The final dataset consists of daily runoff time series aggregated from the $500 \times 500\,\mathrm{m}$ PREVAH grid to the 307
projection catchments used in this study, provided for each model chain. Hydro-CH2018 thus serves as an operational reference
for Swiss-scale hydrological impact assessments (Brunner et al., 2019a).





## 3 Methods

### 3.1 LSTM models

Long Short-Term Memory (LSTM) networks are a class of recurrent neural networks (RNNs) specifically designed to capture

long-range dependencies in sequential data by addressing the vanishing and exploding gradient problem, whereby gradients become too small or too large during backpropagation through time, preventing the model from learning long-term patterns (Hochreiter and Schmidhuber, 1997). An LSTM cell comprises a memory cell and three trainable gates – input, forget, and output – that dynamically regulate information flow across time steps, allowing the model to retain or discard past information depending on its relevance (Gers et al., 2000; Waqas and Humphries, 2024).

This gating mechanism enables LSTMs to maintain internal state memory, making them particularly suitable for problems with delayed or cumulative effects such as snowmelt or soil moisture storage in hydrological systems. Moreover, LSTMs can learn nonlinear, multivariate, and nonstationary input–output relationships directly from data, without requiring explicit knowledge of underlying physical processes. As a result, they have shown particular success in rainfall–runoff modelling, often outperforming traditional conceptual models across diverse climatic regimes (Gauch et al., 2021; Kratzert et al., 2018; Waqas

and Humphries, 2024).

### 3.2 Data pre-processing

All input features, both dynamic and static, were standardized via z-transformation (zero mean, unit variance), based on the training-set statistics. Missing discharge values were masked during training so that only valid observations contributed to loss computation. Glacier area fraction was included as a dynamic input variable, derived from 5-year GloGEMflow simulations

(Zekollari et al., 2019) and aggregated to the catchment scale. Values were linearly interpolated to annual resolution to obtain annual glacier fractions for each catchment and climate chain. This represents a key deviation from Kraft et al. (2024), where glacier fraction was held constant in time (see Sect. 2.5 for glacier dataset description).

More specifically, the dataset was processed as follows:

– For training (1961–2024, 96 observational catchments):

– a constant glacier extent corresponding to the 1990 value was assumed for 1961–1990, supported by observations suggesting limited glacier change over this period (Huss et al., 2010).

– from 1990–2015, we used the simulation-based glacier area from the historical E-OBS-driven run.

– for 2015–2024, we applied the mean of the 30 model chains to approximate recent glacier evolution.

– For projections (1981–2100, 307 projection catchments):

– we used the chain-specific gridded glacier projections corresponding to the CH2018 chains retained for runoff simulations, aggregated to the 307 catchments of Hydro-CH2018 and our LSTM runs.





### 3.3 Model architecture and adaptations

We adopt the same core architecture as Kraft et al. (2024): an LSTM-based model that predicts daily runoff from sequential meteorological inputs and static catchment attributes. The architecture consists of a single temporal LSTM layer, followed by a fully connected output layer that maps the temporal encoding to a scalar runoff prediction for each day. Our configuration corresponds to the model setup identified by Kraft et al. (2024) as their best-performing configuration among 24 tested alternatives (see their Sect. 3.6). In this configuration, static and dynamic inputs are first mapped to the same dimensionality, and then combined via broadcasting in time, followed by addition of their encodings. This early fusion allows catchment-specific information to directly inform the temporal encoding, thereby improving model generalization across heterogeneous hydroclimatic regimes. A full technical specification of the architecture, available hyperparameters, and optimization details is provided in the original publication. Small adaptations, e.g., taking glacier fraction as an additional dynamic input, were made here to better accommodate our study objective.

### 3.4 Training and optimization

The model was trained using backpropagation (Amari, 1993), which updates internal parameters to minimize the difference between predicted and observed runoff. The optimization objective was the mean squared error (MSE) between normalized simulated and observed discharge (Kraft et al., 2024). Parameter updates were performed using the AdamW optimizer (Loshchilov and Hutter, 2019), which adapts learning rates per parameter and applies decoupled weight decay for regularization. To mitigate overfitting, early stopping was applied based on validation loss: training halted once validation performance no longer improved for a given number of epochs, and the best model (in terms of validation MSE) was retained for prediction on the test set (Kraft et al., 2024).

Hyperparameter tuning was performed in this study using the Optuna framework (Akiba et al., 2019) during the first cross-validation iteration (see next section), following the same procedure as Kraft et al. (2024). The final set of parameters and search spaces are provided in Table A2.

### 3.5 Cross-validation strategy

We applied an 8-fold spatial cross-validation scheme to assess generalization across catchments. The 96 low-impact Swiss catchments were randomly partitioned into eight folds; in each iteration, six folds were used for training, one for validation, and one for testing. A fixed temporal split was applied within each fold: 1961–2015 for training, 2016–2018 for validation, 2019–2023 for testing.

This two-dimensional splitting (spatial × temporal) guards against both spatial overfitting and temporal leakage. The model was retrained independently in each fold, resulting in eight sets of trained weights used for subsequent evaluation and projection. This setup therefore allows the best use of the available data for model training.





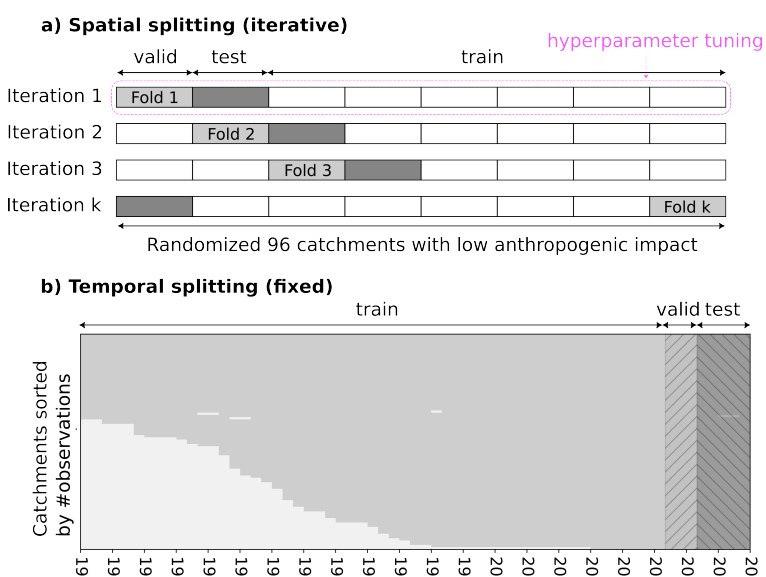

**Figure 2.** Cross-validation design for LSTM training on observed runoff. (a) Spatial splitting: 96 catchments with minimal anthropogenic influence are randomly partitioned into $k = 8$ folds; in each iteration, one fold serves as test, a second as validation, and the remaining six as training. (b) Temporal splitting: a fixed chronology partitions data into training, validation, and test segments. Panels (a) and (b) together define the data subsets used in each cross-validation round. Hyperparameters were optimized during the first iteration, with the resulting settings carried over for all subsequent cross-validation runs. Adapted from Kraft et al. (2024).

## 3.6 Application to climate projections

The trained LSTM model was applied to climate projections spanning 1981–2100, using the CH2018 daily catchment-averaged meteorological forcing data, together with glacier projections and static catchment attributes. The projection inputs were com-
bined into a single dataset. For each cross-validation fold, variables were normalized as $(x - \text{mean})/\text{std}$ using the mean and standard deviation from that fold's training subset. This normalization was applied at runtime.

For the projection experiments, we used the 14 CH2018 model chains available under RCP8.5 (Table A1), representing the intersection of chains for which atmospheric, hydrological (Hydro-CH2018), and glacier evolution data are jointly available. When a chain existed at both EUR11 and EUR44 resolution, the higher EUR11 resolution was selected.
For each chain, the eight LSTM models obtained from the cross-validation folds were run independently, and the daily median across these ensemble members was taken as the final prediction.




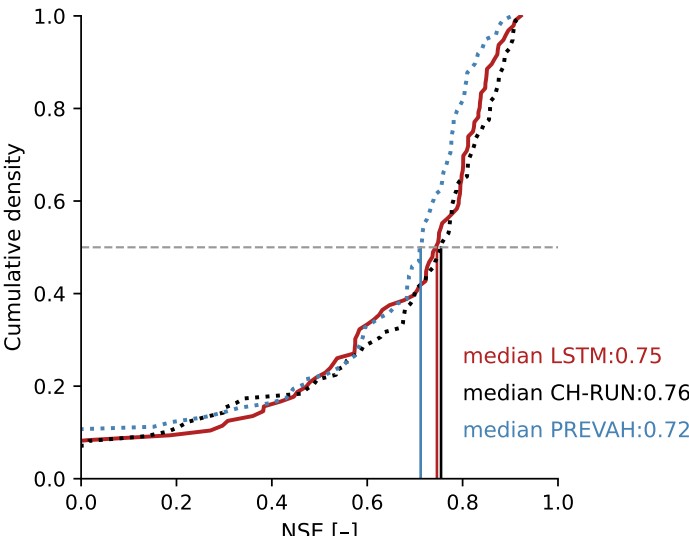

**Figure 3.** Empirical cumulative density functions of Nash–Sutcliffe Efficiency (NSE) across test catchments. Solid red shows the CDF for our LSTM on the test set; dotted black and dotted blue show the CDFs for the CH-RUN best LSTM and the PREVAH benchmark, respectively, both taken from Kraft et al. (2024). Vertical lines mark the median NSE of each distribution; our model (solid red), CH-RUN (black), and PREVAH (blue). Higher NSE indicates better agreement between simulated and observed runoff in the observational catchments.

## 4 Results

### 4.1 Model performance on the test set

Figure 3 presents empirical cumulative density functions (CDFs) of test-set Nash–Sutcliffe Efficiency (NSE) for our LSTM
(solid red), the CH-RUN LSTM (dotted black) and the PREVAH benchmark (dotted blue). Vertical lines indicate each model's median NSE, facilitating a direct skill comparison. In this first validation step, the LSTM achieved a median NSE of 0.75 across the 96 observational catchments, closely aligned with the original CH-RUN setup (0.76; (Kraft et al., 2024)). This performance is also notably higher than the PREVAH benchmark, which reached 0.72 on similar catchments.

### 4.2 Dynamical stability of runoff projections

Having established baseline performance, we next assess the dynamical stability of projections, asking whether the two models respond consistently across climate chains and time scales. Figure 4 shows the Pearson correlation coefficients between LSTM and PREVAH runoff simulations for each of the 14 RCP 8.5 ensemble chains, computed separately for two time horizons (1991–2020 and 2071–2099) and two temporal aggregations (daily and annual).

Overall, correlation patterns are consistent across chains and horizons, highlighting that both models simulate runoff dynam-
ics very similarly. Annual correlations are highest, with medians near 0.95 and narrow interquartile ranges for both horizons




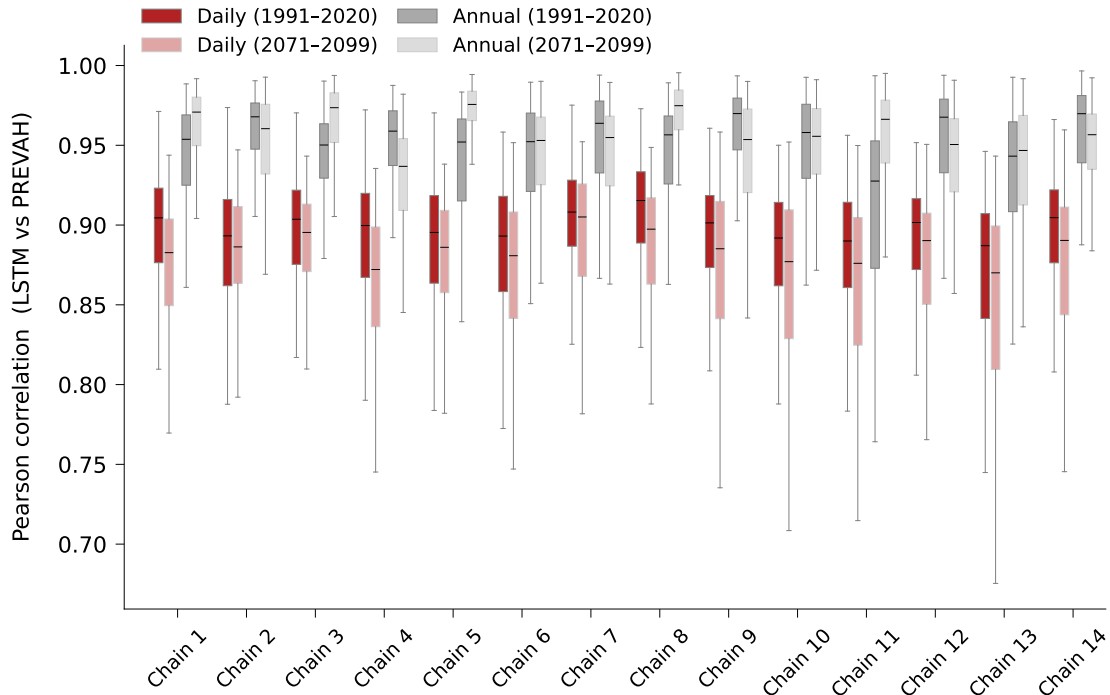

**Figure 4.** Pearson correlations between LSTM and PREVAH runoff under RCP8.5 scenario for each of the 14 ensemble chains (chain numbering corresponds to Table A1). For each chain, four boxplots display per-catchment correlation coefficients: daily (1991–2020 solid red; 2071–2099 transparent red) and interannual (1991–2020 solid grey; 2071–2099 transparent grey). The spread of each boxplot reflects variability in correlation across the catchments. Higher correlation values indicate stronger agreement between LSTM projections and PREVAH reference simulations.

relative to daily correlations, showing no systematic change between past and future. Daily correlations remain strong (∼0.9) but decline slightly and broaden in the future period. These results provide a first indication of projection stability, before turning to spatial patterns of runoff change.

### 4.3 Historical baseline – spatial patterns

At the annual scale, both models show strong agreement in spatial patterns and magnitudes. This is demonstrated in Figure 5, which maps mean seasonal runoff for 1991–2020, showing the LSTM median, the PREVAH median across all chains, and their difference. We use 1991–2020 as the historical baseline, consistent with the reference periods adopted in the most recent Swiss climate scenario assessment CH2025 (MeteoSwiss and ETH Zurich, 2025). This provides a recent and coherent climatological benchmark for comparing the LSTM and PREVAH model responses.

Results are presented for three characteristic periods: Annual values integrate the full water balance and long-term trends, Winter (DJF) highlights snow accumulation and melt dynamics, and Summer (JJA) captures low-flow and potential drought





**Figure 5.** Spatial distribution of mean seasonal runoff for the RCP8.5 historical baseline (1991–2020). Panels are arranged in a 3×3 grid by season (rows: Annual, Winter [DJF], Summer [JJA]) and by model output (columns: LSTM median, PREVAH median, $\Delta$ = LSTM–PREVAH). Each map shows the median over 14 ensemble chains of the 30-year mean seasonal total. Colorbars under columns 1–2 span the 5th–95th percentile of median runoff (mm period$^{-1}$), and under column 3 a diverging scale for $\Delta$ (mm period$^{-1}$). Hatch marks on the LSTM and PREVAH maps denote catchments whose coefficient of variation of the long-term mean seasonal runoff across ensemble chains exceeds 10%. Hatch marks in column 3 flag catchments where a robust signal-to-noise ratio, $\text{SNR} = |\text{median}(r_c)|/\text{MAD}(r_c)$ with $r_c = 2(L - P)/(|L| + |P| + \varepsilon)$, is $< 1$, indicating that the median model difference is small or not robust relative to inter-chain variability.





conditions. For spatial reference, the Swiss biogeographical regions are used (Figure A1; BAFU (2022)). Both models show high runoff on the northern and southern flanks of the Alps and in the western central Alps, and low values on the Plateau. A slight increase is visible in the Jura. Divergences arise where the LSTM does not reach PREVAH's magnitude, notably in
southern and central Switzerland, the eastern Alps, and parts of the west. Overall, the LSTM projects slightly lower annual runoff, and almost no catchments are hatched, indicating low inter-chain variability.

In winter, patterns remain similar but divergences increase. Both models show a northward shift of higher runoff and lower totals in the Alps and southern flank. Yet across much of Switzerland, particularly the Plateau, Jura, and western/eastern regions, LSTM values are lower, as seen in the pinkish difference maps. Both models show larger inter-chain variability in
alpine and southern regions (hatched catchments), reflecting greater uncertainty in low-flow conditions.

In summer, the models again agree well, though difference maps show pronounced contrasts. This largely reflects the much higher absolute runoff magnitudes in summer compared to winter. Both capture alpine peaks and low Plateau/Jura runoff, but the LSTM tends to project higher values in the western central Alps and northern flank, while PREVAH simulates higher values in the eastern Alps and southern flank. Inter-chain variability is larger for PREVAH on the Plateau and in alpine areas, whereas
the LSTM shows such variability only locally.

Across all seasons, hatching in the $\Delta$-panels marks catchments with low inter-model robustness, where the LSTM-PREVAH median difference is small relative to inter-chain variability. It occurs only sporadically and without a clear spatial pattern, mostly coinciding with catchments showing weak inter-model differences. With the baseline established, we next turn to projected changes towards the late 21st century.

### 280   4.4   Future projections – spatial patterns

The projections show strong consistency between models in space and magnitude: both simulate overall annual drying, wetter winters, and pronounced summer drying across nearly all catchments. Figure 6 maps seasonal runoff changes for 2071–2100 relative to 1991–2020 under RCP 8.5, showing the ensemble-median change from the LSTM, from PREVAH, and their difference. These maps highlight agreement and divergence in projected end-century shifts.

At the annual scale, the LSTM generally predicts slightly stronger drying, especially over the Plateau, northern pre-Alps, and western central Alps. PREVAH shows the same pattern with less intensity. Notable catchment-specific differences include one glacier-rich basin where the LSTM projects positive change. While the LSTM tends to predict stronger drying overall, inter-model differences vary in sign in the central Alps.

In winter, both models consistently project wetter conditions, strongest in alpine regions. Divergences are also largest there,
with PREVAH simulating slightly larger increases. Plateau changes are negligible in both models. Difference maps show LSTM increases as systematically but only slightly weaker than PREVAH's.

In summer, both models agree on strong drying, most pronounced in the Alps and southern flank, with more moderate changes on the Plateau and in the Jura. PREVAH generally simulates stronger drying, particularly in alpine regions, though a few northern Alpine catchments show the reverse, with larger LSTM declines.





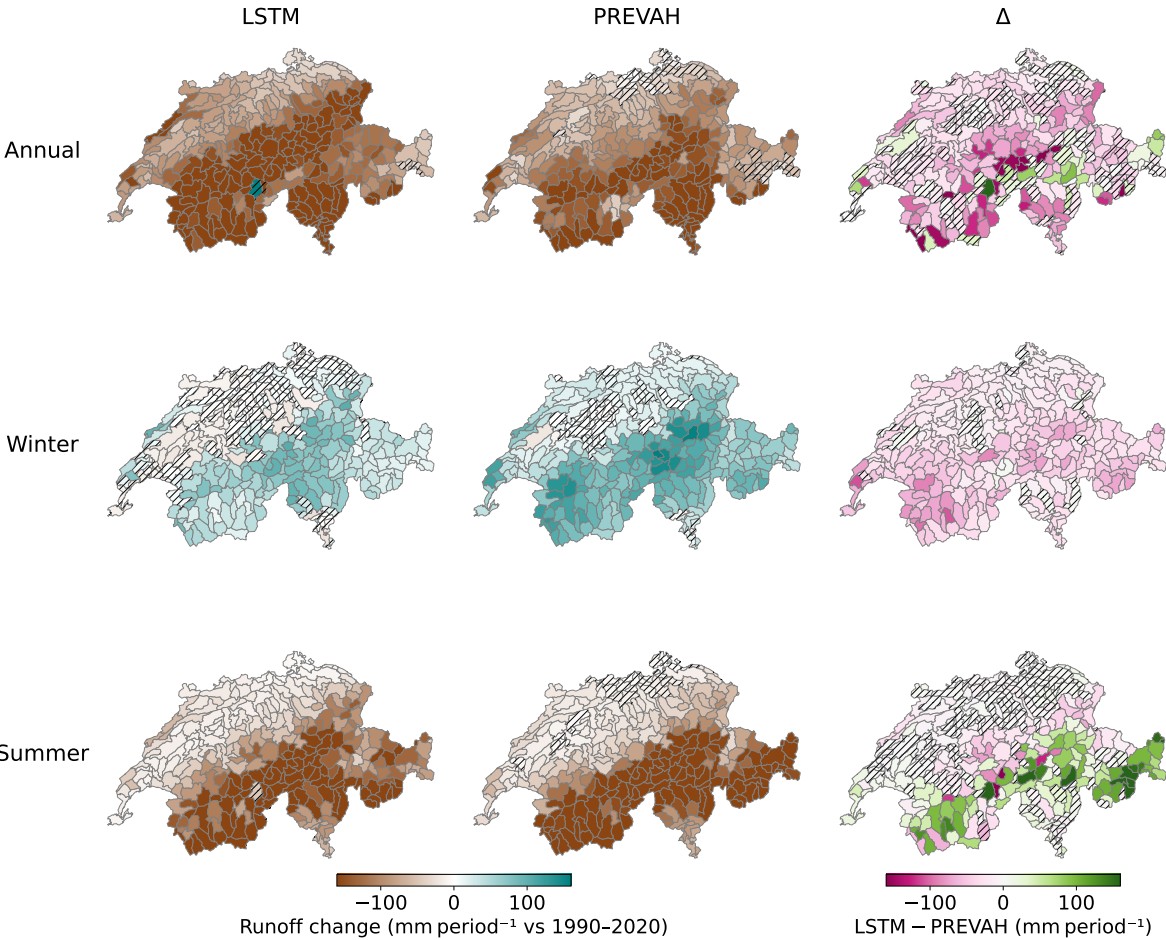

**Figure 6.** Spatial distribution of median seasonal runoff changes for 2071–2100 relative to the 1991–2020 reference period under RCP8.5. Panels are arranged in a 3×3 grid by season (rows; Annual, Winter [DJF], Summer [JJA]) and by model output (columns; LSTM median, PREVAH median, $\Delta$ = LSTM–PREVAH). Columns 1–2 show the change averaged first over 30 years and then taking the median across 14 ensemble chains; column 3 shows the difference between those two median changes. Hatch marks in columns 1–2 flag catchments where the signal-to-noise ratio of the ensemble, $\mathrm{SNR} = |\mathrm{median}(\Delta)|/\mathrm{MAD}(\Delta)$, is $< 1$, indicating that ensemble spread is large relative to the median signal. Hatch marks in column 3 flag catchments where a robust signal-to-noise ratio, $\mathrm{SNR} = |\mathrm{median}(r_c)|/\mathrm{MAD}(r_c)$ with $r_c = 2(L - P)/(|L| + |P| + \varepsilon)$, is $< 1$, indicating that the median model difference is small or not robust relative to inter-chain variability.





Across all seasons, within-model signals are generally robust, with few catchments where ensemble spread rivals the median change (hatched catchments). Exceptions occur in winter over the Plateau, especially for LSTM, where ensemble spread is large relative to weak signals. Catchments with low inter-model robustness — as indicated by hatching in the Δ-panels — are scattered and somewhat more common in summer, especially over the Plateau and the southern flank of the Alps. Having established national-scale changes, we next turn to representative catchments.

## 4.5 Case studies

Overall, PREVAH and LSTM projections align well for selected locations. Figure 7 shows RCP 8.5 runoff-change trajectories (relative to 1991–2020) for six regime-representative catchments. Thin lines denote individual ensemble chains (red = LSTM, blue = PREVAH), thick curves the chain medians, and grey lines their difference. Results are shown for Annual, Winter, and Summer. Overall, median trajectories agree well across seasons and regimes.

At the annual scale, all catchments show negative changes by 2100, though magnitudes differ. Discrepancies are largest in the highly glaciated catchment, where the difference curve departs strongly from zero: PREVAH projects stronger mid-century drying, but the LSTM surpasses it by 2100. Together with the southern Alpine catchment, this represents the strongest decline. In contrast, slightly glaciated catchment show closer agreement with nearly overlapping median trajectories, while high-alpine snow and pre-alpine rain–snow regimes exhibit stronger LSTM drying toward the end of the century. The pre-alpine case shows a systematic median shift in the second half of the century, despite similar trends. The lowland rain regime shows smaller declines and tight agreement, whereas the southern Alpine basin displays stronger negative changes in both models, with the LSTM consistently drier. Despite variations, both models agree on the sign of change across all regimes by 2100. Individual chains show substantial scatter at annual scale, and not all agree on the sign within the same ensemble.

In winter, changes are weaker and opposite in sign, with all catchments showing positive trends by 2100. Increases emerge steadily after ~2030. Differences are again largest for the highly glaciated basin, but also notable in the lowland rain catchment, where PREVAH projects stronger late-century increases. More generally, PREVAH simulates larger positive changes in snow-rich zones, while the LSTM response is more moderate.

In summer, declines begin as early as 2020–2030, especially at medium to high elevations. Summer drying is the most consistent and severe change, exceeding winter increases. For the highly glaciated basin, the summer decline nearly matches the annual decrease. Importantly, both models agree closely on the summer trajectories, with consistent magnitude and timing across all regimes.

Because Swiss hydrological regimes are strongly tied to elevation (Muelchi et al., 2021a), the next section examines results aggregated by altitude bands.

## 4.6 Elevation influence

The models agree well in terms of seasonal elevation-dependent runoff change patterns, although the magnitude of changes differs. Figure 8 displays summaries of 2071-2100 seasonal runoff changes (relative to 1991–2020), grouped into six elevation bands and contrasted between LSTM (red) and PREVAH (blue).





**Figure 7.** Chain-wise runoff seasonal change time series for six representative catchments: Rosegbach–Pontresina (highly glaciated), Kander-Hondrich (slightly glaciated), Plessur–Chur (high-Alpine snow), Emme–Emmenmatt (pre-Alpine rain/snow), Venoge–Ecublens (lowland rain), and Verzasca–Lavertezzo (southern Alpine), selected following Muelchi et al. (2021a) and matched to the 307 projection catchments (IDs in parentheses, locations shown in Figure 1). Panels are arranged by catchment regime (rows a–f) and season (columns: Annual, Winter, Summer). Runoff changes (mm period$^{-1}$) were computed per chain as deviations from the 1991–2020 reference mean, area-weighted across projection catchments to the observational boundary (when more than one catchment), then smoothed with a Gaussian filter ($\sigma = 2$ yr). Thin lines show individual RCP8.5 ensemble chain changes (red: LSTM; blue: PREVAH), thick solid lines their chain-median changes, and gray lines the difference (LSTM - PREVAH).



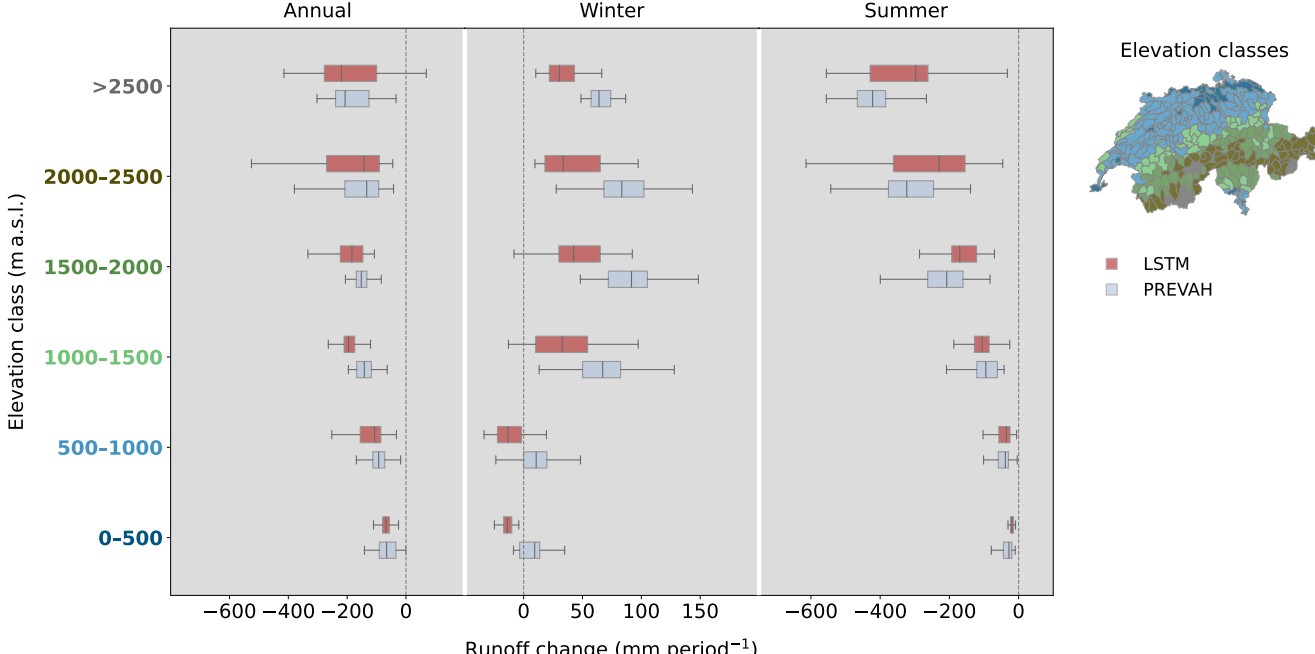

**Figure 8.** Distribution of seasonal runoff changes (mm period$^{-1}$ vs 1991–2020) for 2071-2100 under RCP8.5, aggregated by elevation class. Panels correspond to seasons (Annual, Winter [DJF], Summer[JJA]). Within each panel, boxplots show catchment-level changes: red for LSTM, blue for PREVAH. Runoff changes were computed first per catchment (mean over years then median over the 14 ensemble chains), then grouped by elevation bands. The inset map at right delineates the six elevation bands, and y-axis tick labels are colored to match these bands.

At the annual scale, the two models agree well on both sign and magnitude. Negative changes intensify with altitude, showing stronger drying at higher elevations. The LSTM tends to exhibit larger variability – i.e., wider distributions – especially in the highest bands, but median values remain similar, so neither model consistently projects stronger or weaker changes. When variability across catchments is considered, the LSTM leans toward slightly larger negative shifts.

In winter, the elevation signal is less regular. At low elevations (0–1000 m), the models even diverge in sign, with PREVAH showing a slight positive change and the LSTM a weak negative one, though both values are close to zero. At higher elevations, both indicate positive changes that increase to a maximum around 1500–2000 m and then decrease – while remaining positive – above 2000 m, yielding a characteristic D-shaped response. PREVAH simulates stronger increases across all bands, while the LSTM shows a more heterogeneous response.

In summer, elevation dependence is most pronounced. Both models project drying that increases almost linearly with altitude. Changes remain modest below 1500 m but intensify sharply above, reaching values comparable to annual changes. PREVAH generally simulates slightly stronger declines, while the LSTM again shows wider spreads, especially at mid to high elevations.



Having examined seasonal and elevation-dependent changes, we now turn to runoff extremes, where LSTMs are known to face limitations.

### 4.7 Extreme representation

In alignment with previous studies (e.g., Baste et al., 2025), our LSTM model shows a different behaviour than the reference model PREVAH for extreme runoff values. Figure 9 summarizes national-scale patterns of annual runoff extremes under RCP8.5 for 2071–2099, displaying maxima (blue) and minima (red) from the LSTM and PREVAH, together with their differences.

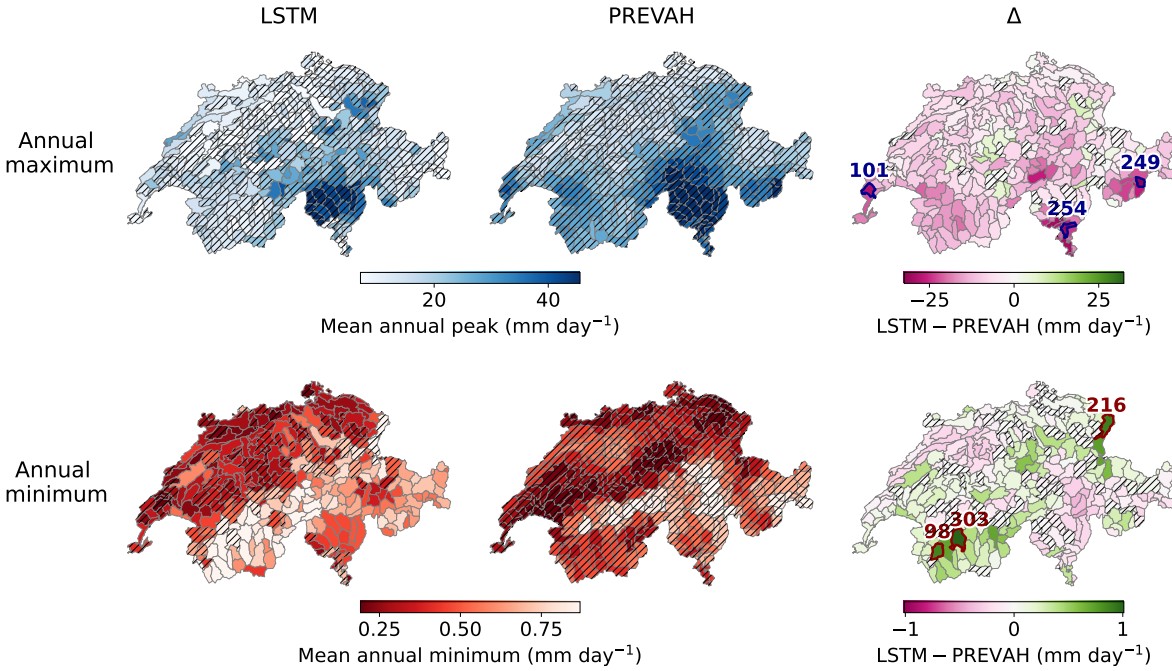

**Figure 9.** Spatial distribution of annual maximum and minimum runoff for 2071–2099 under RCP8.5. Panels are arranged in a 2×3 grid: the top row shows mean annual maxima, and the bottom row mean annual minima; columns show the LSTM median (left), the PREVAH median (center), and their difference $\Delta$ = LSTM − PREVAH (right). Annual maxima (minima) are defined as the maximum (minimum) daily runoff per chain and year, then averaged over 2071–2099 and the median taken across the 14 ensemble chains. Colorbars under columns 1–2 span the 5th–95th percentile of model medians (mm day$^{-1}$), while column 3 uses a diverging scale for $\Delta$ (mm day$^{-1}$). Hatch marks on the LSTM and PREVAH maps denote catchments whose coefficient of variation of the mean annual maxima/minima across ensemble chains exceeds 10%. Hatch marks in column 3 flag catchments where a robust signal-to-noise ratio, $\mathrm{SNR} = |\mathrm{median}(r_c)|/\mathrm{MAD}(r_c)$ with $r_c = 2(L-P)/(|L|+|P|+\varepsilon)$, is $< 1$, indicating that the median model difference is small or not robust relative to inter-chain variability. Three catchments outlined in dark blue and dark red on the difference maps have the largest LSTM–PREVAH discrepancies and are selected for further investigation.





Both models reproduce a similar spatial structure for annual maxima, with the largest values in the central Alps and on the southern flank. Clear magnitude differences emerge nevertheless: PREVAH systematically simulates higher peaks, particularly in the western and eastern central Alps and southern flank, with further discrepancies in parts of western Switzerland. The
difference map shows a near-continuous underestimation by the LSTM, visible in widespread pink shading. Inter-chain variability is large in both models and slightly larger for PREVAH. Locations of low inter-model robustness (hatched) are rare and show no clear spatial structure.

For annual minima, spatial patterns are consistent across models, with the lowest values on the Plateau and comparatively higher minima in alpine regions. The LSTM projects less severe minima than PREVAH, especially in the western central Alps
and along the northern flank, visible as green shading in the difference map. Inter-chain variability is comparatively small for the LSTM (mainly on the Plateau) but widespread for PREVAH across Switzerland. Low-robustness differences occur slightly more often than for maxima but remain spatially sporadic.

Figure 10 further explores these divergences in catchments with the largest differences by showing how wet and dry precipitation events are translated into runoff.

For maxima, PREVAH responds much stronger to 3-day precipitation sums, with steeper runoff–precipitation slopes, especially in alpine catchments (249, 254). This is evident from the scatter plots and quadratic fits. For minima, both models show weak links to 21-day precipitation deficits, but the LSTM consistently simulates higher baseflows, evident as a systematic vertical offset between fitted curves. Here, the key difference lies in magnitude rather than sensitivity. Together, these results highlight that PREVAH exhibits stronger sensitivity to precipitation extremes, whereas the LSTM shows a regression-to-the-
mean behaviour, dampening both high and low flows. This consistent response pattern informs the model limitations discussed in Section 5.7.

## 5 Discussion

### 5.1 Model calibration

As expected, our LSTM model achieves performance closely aligned with the original CH-RUN implementation (Kraft et al.,
2024). The results confirm that our adaptations – most notably the integration of a dynamic glacier fraction and a different temporal split in the training scheme – do not compromise reconstruction skill, which remains consistent with Kraft et al. (2024). This strong agreement establishes a reliable baseline for subsequent projection analyses, while also emphasizing that present-day skill does not automatically guarantee robustness under future climate conditions (Wi and Steinschneider, 2022; Natel De Moura et al., 2022). Nonetheless, the high reconstruction accuracy rules out calibration error as a major source of
later discrepancies, strengthening confidence in the interpretability of projection results.




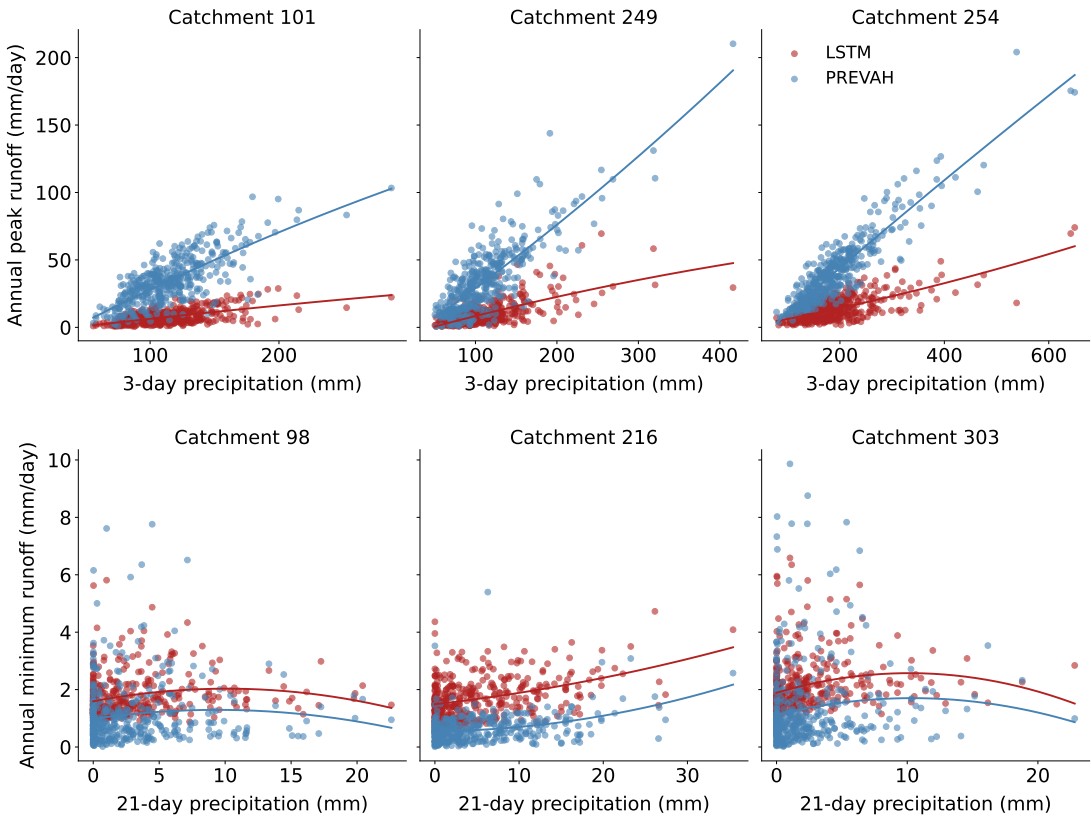

**Figure 10.** Catchment-level relationships between antecedent precipitation and extreme runoff under RCP8.5 for 2071–2099. Top row: annual maxima (red: LSTM, blue: PREVAH) at the date of maximum 3-day rolling precipitation versus that precipitation sum, for the three catchments with the largest negative LSTM–PREVAH maxima differences (IDs 101, 249, 254). Bottom row: annual minima runoff at the date of minimum 21-day rolling precipitation versus that precipitation sum, for the three catchments with the largest positive LSTM–PREVAH minima differences (IDs 98, 216, 303). Each point is one chain–year event; quadratic regression curves are overlaid.

## 5.2 Dynamical stability of runoff projections

As a first step in assessing projection consistency, we examined catchment-wise correlations between LSTM and PREVAH across the RCM ensemble at daily and annual scales (see Section 4.2, Figure 4). Correlations remain high, particularly at the annual scale, indicating a high degree of consistency between the two models even under changing climate conditions. However, annual correlations mainly reflect agreement in the direction of change rather than its magnitude, since the LSTM may underestimate runoff in some cases (e.g., saturation effects; see Baste et al. (2025)), an aspect examined later. As expected, annual correlations exceed daily ones, as temporal aggregation smooths high-frequency variability and timing mismatches, isolating the climate-driven signal. The limited spread and regularity across ensemble members at both scales further shows that





the LSTM response is consistent regardless of the specific climate forcing, suggesting that variability introduced by different
RCM chains is minor compared to overall model skill.

Together, these findings provide a first indication of the model's stability and its capacity to generalize from historical training to long-term projections, particularly in maintaining consistent interannual runoff dynamics across Switzerland. This dynamical consistency establishes a solid basis for examining where spatial patterns diverge, and why.

## 5.3 Historical baseline – spatial patterns

We first verify that the LSTM delivers spatially stable runoff fields under present-day forcing (Figure 5), before moving to altered climates. At the annual scale, the near-absence of hatching reflects that larger mean flows buffer chain-to-chain variability (columns one and two).

In winter, Alpine means align closely, indicating that both models are consistent in reproducing snow- and ice-controlled runoff despite the LSTM lacking an explicit cryospheric module (Kratzert et al., 2019b). This robustness supports confidence
in its ability to capture future cold-season dynamics. Differences over the Plateau likely reflect rain–snow partitioning, with PREVAH routing more liquid water directly to runoff. Winter hatching in columns 1 and 2 highlights greater relative uncertainty in low-flow settings, but its presence in both models reinforces signal consistency.

In summer, high alpine runoff is consistent with snow and ice melt dynamics: despite elevated evapotranspiration, meltwater sustains strong flows, while mid-elevation catchments release water earlier in spring and lower-elevation ones remain dry with
little or no accumulation (Engel et al., 2015; Muelchi et al., 2021a; Jenicek et al., 2016). Across much of the Alps, the LSTM produces slightly higher values, likely because PREVAH releases part of the melt earlier, leaving less for July–August. In the eastern central Alps, the LSTM yields lower flows, possibly reflecting different learned associations with precipitation, glacier melt, or evapotranspiration. Both models capture the very low Plateau and Jura totals, with residual biases potentially linked to LSTM-learned evapotranspiration patterns.

Overall, both models remain consistent for the historical seasonal baseline. Summer differences point to subtle variations in how snow and ice storage and melt are represented, rather than a fundamental LSTM instability. PREVAH also shows a broader spread in summer (not only where median values are low, see hatching in column two, third row), suggesting less constrained projections, while the LSTM maintains tighter consistency across chains.

Finally, the Δ-panels (Figure 5, third column) add ensemble context to inter-model differences, with hatching marking
catchments where the LSTM–PREVAH median difference is small or not robust relative to inter-chain variability; its scarcity indicates that most differences between the two models are spatially systematic rather than random across chains.

## 5.4 Future projections – spatial patterns

Having confirmed stability under present-day conditions, we now examine end-century (2071–2100, RCP 8.5) change maps to test whether the LSTM extrapolates spatial signals consistently under strongly altered forcing (Figure 6).

At the annual scale, drying is broadly consistent across Switzerland, with one notable exception: a glacier-rich catchment where the LSTM projects a positive change while PREVAH shows depletion. This may reflect retained meltwater contributions



in the LSTM versus earlier glacier exhaustion in PREVAH (Huss and Hock, 2018; Zekollari et al., 2019), though it could also reflect an LSTM artefact. More generally, mismatches cluster in alpine regions, where runoff volumes are highest and hydro-related processes such as rain–snow partitioning, melt rates, and storage release are most complex (FOEN, 2021; Hock et al., 2019; Gurtz et al., 2003). Differences may stem from assumptions about the pace of glacier regime shift, with the LSTM implying a faster transition and thus steeper annual declines.

In winter, both models project wetter conditions, strongest in the Alps, consistent with reduced cryospheric storage capacity and a shift from snowfall to rainfall (FOEN, 2021; Muelchi et al., 2021a; Kotlarski et al., 2022). PREVAH changes are systematically larger, likely because its physics convert a greater fraction of snowfall into immediate runoff. The LSTM may understate increases in catchments where rain-over-snow transitions exceed its training range (Wi and Steinschneider, 2024). Plateau catchments remain near-zero in both models, being less sensitive to snow and glacier dynamics, and precipitation phase shift.

In summer, projections align with expectations: reduced storage and accelerated melt accentuate water deficits, especially in the Alps and southern flank (Muelchi et al., 2021; Brunner et al., 2019b). PREVAH generally produces larger declines, consistent with its wetter winters and earlier snow/ice release, while LSTM drying is weaker, possibly from preserved winter storage. Despite this amplitude shift, possibly traced to winter divergence, spatial patterns are consistent, with no indication of LSTM instability.

PREVAH's larger winter increases likely explain its stronger summer drying. Interestingly, however, the LSTM amplifies the annual drying signal. Timing shifts alone, such as delayed snowmelt or prolonged winter storage, cannot explain this, since they would not yield a more negative annual change. A more plausible explanation is that the LSTM produces less winter runoff, offsetting weaker summer drying, or conversely that PREVAH's wetter winters balance its drier summers. The LSTM may therefore redistribute changes more evenly, producing a net annual reduction. Such behaviour may reflect its limited training exposure to extreme seasonal transitions.

The largest summer mismatches occur in the Alps (see Figure 6, third row), Switzerland's key snow- and ice-fed water reservoirs, highlighting the need for close scrutiny of these basins (NCCS, 2021a; Brunner et al., 2019b). Still, the scarcity of hatching across change maps (columns 1-2) indicates robust within-model signals, with ensemble spread smaller than the signal itself. In the $\Delta$-panels (Figure 6, third column), hatching is sparse—mainly in summer over the Plateau and the southern Alps—indicating that, in most regions, differences between the LSTM and PREVAH median anomalies are consistent across chains. Where hatching appears, inter-model contrasts are small or not robust.

It is worth noting that EURO-CORDEX ensembles, underlying the CH2018 scenarios, tend to underestimate summer warming and drying because many RCMs neglect aerosol decline and the associated shortwave increase (Schumacher et al., 2024). Runoff changes reported here may thus be conservative relative to more recent scenarios.

Overall, across both the 1991–2020 baseline and the RCP8.5 end-century change maps, the LSTM is consistent with PREVAH in reproducing Switzerland-wide change patterns, confirming that a data-driven model can generalize its learned hydro-climate relationships to out-of-distribution forcing, despite a few mismatches discussed below (Kratzert et al., 2019a; Natel De Moura et al., 2022).





## 5.5 Case studies

With chain-by-chain and then large-scale stability established, we next zoom from national maps to six representative catchments that typify Switzerland's runoff regimes (Figure 7). The aim is to test whether LSTM–PREVAH agreement holds at the
catchment level and whether skill diverges in specific regime types. For this, ensemble-median trajectories from both models were compared for annual, winter, and summer runoff.

On an annual scale, drying intensifies after 2040 in slightly and highly glaciated catchments, reflecting the expected post-"peak water" decline (Huss and Hock, 2018). PREVAH appears more buffered in glacier-fed regimes, while the LSTM projects stronger drying by 2100, consistent with national maps.

In winter, both models simulate increasing runoff driven by the shift from snowfall to rainfall, again consistent with national projections (Muelchi et al., 2021a). PREVAH produces larger changes, likely due to its representation of snow-to-rain transitions at lower elevations and immediate routing at higher, snow- and glacier-dominated catchments. LSTM shows a dampened rise, possibly reflecting delayed release of precipitation inputs.

During summer, both models project strong declines in glaciated and alpine catchments, clearly consistent with meltwater
exhaustion (FOEN, 2021). These declines nearly match the annual changes, underscoring summer's dominant role in driving long-term reductions (Hock et al., 2019). Increasing winter runoff does not offset these losses but instead accelerates summer depletion, as seen in Section 5.4. In lowland, southern Alpine, and pre-Alpine catchments, summer declines are likely driven by increased evapotranspiration and reduced soil moisture recharge, though magnitudes remain moderate and may be buffered by enhanced convective precipitation (CH2018, 2018; FOEN, 2021; Kotlarski et al., 2022).

Overall, the LSTM remains consistent with PREVAH across all regimes, with the clearest mismatches confined to glacier-rich and snow-transition catchments. Prior studies show that LSTMs trained on large, diverse datasets can generalize runoff predictions to unseen basins, often outperforming traditional regionalization approaches (Arsenault et al., 2023; Lees et al., 2021; Kratzert et al., 2019b). By leveraging cross-catchment patterns and static descriptors, LSTMs can internalize regime-specific processes and translate climate-forcing changes into realistic runoff responses, even in ungauged basins (Yu et al., 2024;
Wi and Steinschneider, 2022; Kratzert et al., 2024). Our catchment-scale results support this ability, as the LSTM reproduced regime-dependent responses across Swiss catchments (see Figure 7). Since these regimes are strongly tied to elevation, the next section aggregates catchments by elevation bands to test whether these strengths and weaknesses persist at a Switzerland-wide scale (Muelchi et al., 2021a).

## 5.6 Elevation influence

Given the strong link between elevation and hydrological regime in Switzerland, and the earlier finding that LSTM–PREVAH mismatches are most pronounced in glacier-fed and snow-transition catchments, we now assess whether discrepancies scale systematically with altitude (Figure 8). This broader grouping complements the catchment analysis and tests consistency across Switzerland's vertical gradient, crucial for climate adaptation given the role of high-altitude reservoirs (NCCS, 2021a, b).





On an annual basis, both models agree that drying intensifies with elevation. Historically, alpine flows depend strongly on
snow and glacier melt, but under RCP8.5 these sources are depleted by reduced snow accumulation, earlier melt, and long-term
glacier loss (Brunner et al., 2019b; Huss and Hock, 2018). Since winter gains no longer compensate summer losses, annual
changes become increasingly negative at higher altitudes.

In winter, runoff increases across all elevations, strongest at mid-elevations where small temperature shifts cause large
rain–snow transitions, producing the observed "D-shaped" profile with peak increases at 1500–2000 m and weaker signals
above and below. This reflects the present-day zero-degree line (Hock et al., 2019; Beniston, 2003). At low elevations, warm-
ing adds little to already rain-dominated regimes, while at high elevations, temperatures may remain cold enough for much
winter precipitation to persist as snow even under RCP8.5, limiting change. PREVAH systematically projects stronger in-
creases than the LSTM, particularly at mid- to high elevations, likely because it routes rain and snowmelt more immediately
to runoff, whereas the LSTM delays release into spring. These differences mirror the cryospheric process divergences noted at
the catchment scale, now generalized across elevation bands.

In summer, the strongest drying occur in high-elevation catchments ($> 2000$ meters), underscoring the vulnerability of
snow- and glacier-fed regimes (Hock et al., 2019; Zekollari et al., 2019). Drying intensifies with altitude as these flows have
historically been sustained by snow and glacier melt (Huss and Hock, 2018; Brunner et al., 2019b). The LSTM projects
slightly weaker declines than PREVAH and shows a wider spread at the highest bands, likely reflecting both the uncertainty
of extrapolating to rarely observed glacier-fed conditions and its lack of explicit glacier dynamics. Still, both models capture
the same spatial trend and seasonal timing, confirming the LSTM's capacity to reproduce elevation-dependent responses to
climate change.

These elevation-based results extend the catchment-level analysis, confirming that the agreement between LSTM and PRE-
VAH holds beyond individual catchments and can be generalized to groups of catchments sharing similar altitude-dependent
hydrological regimes. Both consistently capture the direction of change, though magnitude differences increase at high eleva-
tions, particularly in winter and summer. As also seen in Figure 6 and discussed in Section 5.4, the LSTM dampens seasonal
contrasts, projecting smaller winter increases and weaker summer decreases, while amplifying drying on an annual basis.

This pattern suggests that simple timing shifts cannot explain the divergence: either the LSTM produces less winter runoff
while PREVAH offsets drier summers with wetter winters, or the LSTM redistributes changes more evenly across seasons.

Across elevation bands, the spread varies systematically with altitude: LSTM projections are wider at mid- to high elevations,
likely reflecting the under-representation of glacier- and snow-fed basins in training (Figure 1) and the absence of explicit
physical constraints, which together inflate variability when extrapolating beyond its core domain (Kratzert et al., 2024).
Stronger climate signals at higher elevations may amplify this sensitivity. By contrast, in lowland rainfall-driven catchments,
well represented in training, LSTM spreads are narrower and sometimes smaller than those of PREVAH.

Overall, vertical stratification of climate impacts is reproduced by both models and aligns with established hydrological
understanding in Switzerland (FOEN, 2021; Muelchi et al., 2021a). Conclusively, both regime-specific and country-wide tests
demonstrate that the LSTM mirrors PREVAH's mean-flow responses in space and time under altered climate, with remaining
challenge at the distribution tails, where LSTMs risk peak-flow saturation (Baste et al., 2025).



## 5.7 Extreme representation

Runoff extremes are critical for applications such as energy production, infrastructure management, and ecosystem protection (Brunner et al., 2019b; Haddad et al., 2025; Caretta et al., 2022), but they remain a recognized challenge for LSTM models due to:

1. saturation of the network state, which limits responsiveness to very large inputs (mechanical limitation; (Baste et al., 2025));

2. scarcity of extreme events in the training record, making them harder to learn (long-tail learning problem; (Martel et al., 2024)); and

3. distribution shifts in climate projections, which generate inputs outside the range seen during training (domain adaptation challenge; (Wi and Steinschneider, 2022; De Silva et al., 2020)).

Comparing LSTM and PREVAH therefore provides valuable insights.

Figure 10 corroborates the maps in Figure 9: the LSTM shows clear signs of peak-flow saturation. PREVAH responds more strongly to heavy precipitation, with a steeper runoff–precipitation slope (Figure 10), while the LSTM's dampened slope supports the interpretation of saturation (even though it does not display the flat saturation plateau sometimes reported in the literature).

From the ensemble perspective (Figure 9), widespread hatching in columns 1–2 indicates strong chain-to-chain sensitivity of peak-flows, and likely nonlinear hydrologic responses during high-flow events. In contrast, hatching in the the $\Delta$panel is scarce, showing that the negative LSTM–PREVAH differences in annual maxima are spatially consistent across ensemble members rather than random. Hence, despite large internal variability, the relative bias between the two models is robust—model formulation matters more than forcing-chain choice.

In contrast, PREVAH tends to simulate deeper deeper lows (Figures 9, 10), with differences that concern magnitude more than sensitivity to precipitation deficits (see Figure 10, bottom row). However, literature suggests that PREVAH may over-dry under high evaporative demand, implying the LSTM could provide a more moderate – and possibly more realistic – baseline representation of future low-flow conditions (Brunner et al., 2019a). Still, this interpretation requires caution.

Ensemble patterns also diverge: PREVAH shows extensive hatching, reflecting greater chain-to-chain variability, while the LSTM exhibits sparser hatching, likely due to stronger observational regularization and less ability to generate divergent trajectories. Importantly, a broader spread for PREVAH should not be conflated with lower realism, given the challenges of simulating low flows under strong warming. In the $\Delta$panel (Figure 9), hatching appears slightly more often for minima but remains spatially sporadic. This pattern indicates that low-flow differences between LSTM and PREVAH are smaller in magnitude and less robust across chains, reflecting greater sensitivity to chain-specific conditions than for maxima.

Extremes therefore expose the main boundary of LSTM projections: systematic peak-flow saturation leading to muted maxima relative to a process model. This behavior is consistent with prior findings and unlikely to result from PREVAH overestimating wet extremes (Baste et al., 2025; Wi and Steinschneider, 2024).





## 5.8 Strengths and limitations of the LSTM approach

LSTM-based models offer several advantages over traditional hydrological models. Their computational efficiency enables rapid ensemble simulations, facilitating large-scale scenario testing that can support a broad range of applications, as those highlighted in the Introduction (Fathi et al., 2025). They also learn complex temporal dependencies directly from data and capture cross-basin patterns for built-in regionalization (Kratzert et al., 2019b, a). Unlike process-based models, they also avoid potentially biased parameterizations (Arsenault et al., 2023).

A key limitation, however, lies in their strong dependence on large and high-quality training datasets (Arsenault et al., 2023; Waqas and Humphries, 2024). Evidence shows that LSTMs perform best when trained on diverse, multi-basin records (Kratzert et al., 2024), which are not always available. However, ongoing advances in data acquisition, such as remote sensing and expanding global hydrological databases may gradually alleviate this constraint (Kratzert et al., 2024; Ali et al., 2023). Another challenge is interpretability: pure LSTMs provide little insight into the mechanisms they learn (De la Fuente et al., 2024). As a result and without further complementary experiments, divergences from process-based projections can often only be hypothesized rather than explained. Encouragingly, new interpretability tools reveal internal LSTM representations consistent with hydrological knowledge (Lees et al., 2022; De la Fuente et al., 2024), offering a path toward greater transparency.

One promising avenue to improve both interpretability and robustness include hybrid approaches that combine data-driven efficiency with physical knowledge (Liu et al., 2022; Zhong et al., 2024). These retain the self-learning capacity of LSTMs while enforcing realism through physical constraints, making them at least partially interpretable (Kraft et al., 2022; Waqas and Humphries, 2024). Other promising directions include transfer learning, which improves transferability to unseen climates (He et al., 2023); physics-guided architectures, which use soft constraints to enhance realism (Karpatne et al., 2017; Xie et al., 2021); and the use of improved input features and higher-quality training data, which provide more reliable signals for capturing hydrological processes (Kratzert et al., 2024, 2021; Maity et al., 2024). Collectively, these developments may address current limitations in extreme-event representation, strengthen confidence in projections, and enable more systematic attribution of hydrological responses to specific model components.

## 6 Conclusions

In this study, we evaluated wether an LSTM model can provide robust runoff projections for Switzerland under the CH2018 climate scenarios. The model was trained on historical observations and benchmarked against PREVAH, a widely used process-based hydrological model. Results show that the LSTM delivers stable and spatially consistent projections, with strong agreement to PREVAH across diverse hydrological regimes, elevation, and climate chains. Despite the absence of explicit process formulations for snow and glacier dynamics, the LSTM successfully reproduces key hydrological shifts in both historical and future climates, at the catchment level but also nationally. It consistently captures expected trends across heterogeneous regimes and elevation bands, demonstrating that data-driven approaches can generalize beyond their training domain at country scale.

Nonetheless, limitations remain. Extremes continue to pose challenges, with clear signs of peak-flow saturation constraining the representation of annual maxima. At the same time, the LSTM may offer better performance for low-flow conditions,



585 suggesting that machine learning models may complement process-based models in certain contexts. Altogether, these results underscore both the promise and the current boundaries of LSTMs for climate impact assessment in hydrology.

*Code and data availability.* All data and analysis code required to reproduce the results of this study are provided to the reviewers via a restricted-access Polybox repository. The shared material includes: (i) the processed runoff projection dataset (RCP 8.5 LSTM and PREVAH outputs for 1982–2099), (ii) the 307-catchment shapefile used in the spatial analyses, and (iii) the Jupyter notebook used to generate the

590 figures. The LSTM model implementation itself is not redistributed at this stage. Our model is largely identical to the open-source implementation published in Kraft et al. (2024), available at https://github.com/bask0/mach-flow/. Upon acceptance, we plan to release our model code on a public repository (e.g. GitHub).



# Appendix A:  Additional tables and figures

**Table A1.** List of RCP8.5 chains used in the analysis.

| Chain number | Chain name |
|:---:|:---|
| 1 | CLMCOM_CCLM4_HADGEM_EUR44_RCP85 |
| 2 | CLMCOM_CCLM5_ECEARTH_EUR44_RCP85 |
| 3 | CLMCOM_CCLM5_HADGEM_EUR44_RCP85 |
| 4 | CLMCOM_CCLM5_MIROC_EUR44_RCP85 |
| 5 | CLMCOM_CCLM5_MPIESM_EUR44_RCP85 |
| 6 | DMI_HIRHAM_ECEARTH_EUR11_RCP85 |
| 7 | KNMI_RACMO_ECEARTH_EUR44_RCP85 |
| 8 | KNMI_RACMO_HADGEM_EUR44_RCP85 |
| 9 | SMHI_RCA_CCCMA_EUR44_RCP85 |
| 10 | SMHI_RCA_ECEARTH_EUR11_RCP85 |
| 11 | SMHI_RCA_HADGEM_EUR11_RCP85 |
| 12 | SMHI_RCA_MIROC_EUR44_RCP85 |
| 13 | SMHI_RCA_MPIESM_EUR11_RCP85 |
| 14 | SMHI_RCA_NORESM_EUR44_RCP85 |



**Table A2.** LSTM model parameters and hyperparameter search spaces. The table lists the parameters with their search spaces, final values, and derivation. `model_dim` is the dimensionality of the encoding and hidden state; `enc_dropout` the dropout applied in the encoder prior to the temporal layer. `lr` is the learning rate and `weight_decay` the L2 regularization of the weights. These two are parameters of the optimizer and control the update step size and regularization, respectively (Kraft et al., 2024). `lstm_layers` is the number of stacked temporal layers, `fusion_method` corresponds to the approach described in Sect. 3.2.2, and `criterion` refers to the L2 loss (mean squared error, MSE) used as the training objective. `num_static_in` is the number of static catchment attributes used as input, `num_dynamic_in` the number of meteorological variables, and `num_outputs` the number of predicted target variables, here runoff.

| Parameter | Search space | Final value | Derivation |
|---|---|---|---|
| model_dim | {64, 128, 256} | 256 | tuned |
| enc_dropout | {0.0, 0.2} | 0.2 | tuned |
| lr | {1e–4, 1e–3, 1e–2} | 0.001 | tuned |
| weight_decay | {1e–1, 1e–2, 1e–3} | 0.001 | tuned |
| lstm_layers | | 1 | hard-coded |
| fusion_method | | pre_encoded | hard-coded |
| criterion | | l2 (MSE) | hard-coded |
| num_static_in | | 16 | derived from data |
| num_dynamic_in | | 3 | derived from data |
| num_outputs | | 1 | derived from data |

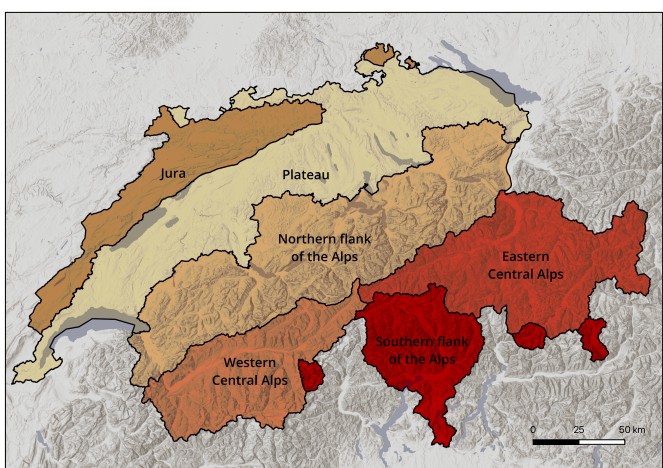

**Figure A1.** The six major biogeographical regions of Switzerland — Jura, Plateau, Northern flank of the Alps, Southern flank of the Alps, Western central Alps, and Eastern central Alps — group areas with similar ecological and biodiversity characteristics based on flora and fauna distribution patterns (BAFU, 2022). In this study, they provide a clear spatial framework to describe and compare results across distinct hydrological contexts.



*Author contributions.* The study was initially framed by LG, BK, and YH, and later fundamentally led by FC. FC took the lead in analyzing the projections,, creation of the visuals, and writing of the manuscript. YH, FC, and BK pre-processed the data, and BK adapted the model and ran the projections. MZ provided the glacier and PREVAH projections. All authors contributed to and approved the final manuscript.

*Competing interests.* The authors declare that they have no conflict of interest.

*Disclaimer.* For transparency purposes, the author acknowledge the use of ChatGPT (version 5.0) to support technical problem-solving during the computational workflow, as well as for assistance with grammar and style refinement. The scientific content, analyses, and conclusions of this work are entirely based on authors' work.

*Acknowledgements.* The authors also acknowledge support from the European Union's Horizon Europe research and innovation program under grant no. 101137682 (AI4PEX—Artificial Intelligence and Machine Learning for Enhanced Representation of Processes and Extremes in Earth System Models). This research was supported by the Swiss Federal Office of Energy as part of the SWEET consortium RECIPE. The authors bear sole responsibility for the conclusions and the results presented in this publication.



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
