# Peer review of "Assessing the stability of LSTM runoff projections in Switzerland under climate scenarios"

_EGUsphere, 2025_

## Referee Comment (RC1)

**Assessing the stability of LSTM runoff projections in Switzerland under climate scenarios**

Review Comments

**Short Summary:**

This study addresses the physical credibility of LSTM predictions for future climate scenarios in Switzerland. The LSTM is trained with observed meteorological and discharge data at the daily resolution for 96 minimally impacted catchments across Switzerland for the period 1961-2024. The glacier area fraction is considered as a dynamic input to account for varying hydrological conditions under climate change and an eight-fold spatial cross validation strategy is adopted. The trained LSTM ensemble is applied to climate projections from 14 CH2018 model chains for the period 1981-2099 and the median of the eight member LSTM ensemble is considered at a daily resolution. The LSTM performance for the test set is slightly better than the process-based model PREVAH used throughout the study as a benchmark.

As a first step to validate projection stability, per-catchment correlation between the LSTM and PREVAH discharge projections is evaluated for each of the 14 projections at a daily and annual scale. This is done for two time periods; 1991-2020 which serves as a historical baseline and 2071-2099. As a next step, the spatial distribution of discharge projects for the historical baseline are assessed at annual and seasonal scales. Finally, the end of century (2071-2099) projections compared to the historical baseline reveal good agreement within the LSTM and the PREVAH with slightly stronger annual drying by LSTM and slightly wetter winters predicted by the PREVAH. The study further analyses these results for six catchments representing diverse hydrological regimes and the LSTM is seen to produce divergent results for the highly glaciated and alpine catchments. The authors then establish a link between discharge projects and the altitude of the catchments, addressing the impact of climate change at varying altitudes and the strong relation between elevation and hydrological regimes in Switzerland. As a last analyses the authors address the representation of extremes by the LSTM and the PREVAH for the climate projects.

The study establishes that the LSTM predictions are spatially consistent with the PREVAH across Switzerland in general. Moreover, the predictions from the two models show similar trends across different hydrological regimes, elevations and climate projection chains. The predictions from the LSTM systematically underpredict the annual maxima in the future scenarios, but could be a better representation of the minima.

**General Comments:**

The study is relevant, well conducted and evaluates the stability of LSTM predictions under climate projections in a systematic manner. The motivation and the objectives for the study are clearly formulated by the authors. The model description as well as the methods for data pre-processing and model training and testing are described clearly. The results are presented in a clear and legible manner, such that every subsection builds on the previous. This is supplemented with adequate figures and descriptive captions. The discussion of the results is well rounded and supported by adequate literature.

As far the technical implementation of the described methods is concerned, I have no major comments.

**Minor/Technical corrections:**

Figure 7: Change legend: Median PREVAH projections are shown in solid blue line, but the legend shows a dashed blue line

L539: 'deeper' repeated